# New Evidence of Significant Association between EBV Presence and Lymphoproliferative Disorders Susceptibility in Patients with Rheumatoid Arthritis: A Systematic Review with Meta-Analysis

**DOI:** 10.3390/v14010115

**Published:** 2022-01-10

**Authors:** Ana Banko, Danijela Miljanovic, Ivana Lazarevic, Ivica Jeremic, Aleksa Despotovic, Milka Grk, Andja Cirkovic

**Affiliations:** 1Institute of Microbiology and Immunology, Faculty of Medicine, University of Belgrade, 11000 Belgrade, Serbia; danijela.karalic@med.bg.ac.rs (D.M.); ivana.lazarevic@med.bg.ac.rs (I.L.); 2Institute of Rheumatology, Faculty of Medicine, University of Belgrade, 11000 Belgrade, Serbia; ivicaje@yahoo.com; 3Faculty of Medicine, University of Belgrade, 11000 Belgrade, Serbia; alexadespotovic21@gmail.com; 4Institute of Human Genetics, Faculty of Medicine, University of Belgrade, 11000 Belgrade, Serbia; milkagrk@gmail.com; 5Institute for Medical Statistics and Informatics, Faculty of Medicine, University of Belgrade, 11000 Belgrade, Serbia; andja.aleksic@gmail.com

**Keywords:** EBV, rheumatoid arthritis, lymphoproliferative disorders, lymphoma, meta-analysis

## Abstract

Development of lymphoproliferative disorders (LPDs) is one of the well-known life-threatening complications in rheumatoid arthritis (RA) patients. However, there is a lack of definitive conclusions regarding the role of Epstein-Barr virus (EBV) activity in RA initiation and progression, especially in promoting LPDs. A systematic review and meta-analysis of studies that reported an EBV positive result in RA-LPD patients and controls were conducted. Studies published before 27 July 2021 were identified through PubMed, Web of Science, and SCOPUS. A total of 79 articles were included in the systematic review. The prevalence of EBV positive result among RA-LPD patients was 54% (OR = 1.54, 95% CI = 1.45–1.64). There was a statistically significant association between EBV presence and LPD susceptibility in RA patients in comparison with all controls (OR = 1.88, 95% CI = 1.29–2.73) and in comparison with LPD patients only (OR = 1.92, 95% CI = 1.15–3.19). This association was not shown in comparison with patients with autoimmune diseases other than RA who developed LPD (OR = 0.79, 95% CI = 0.30–2.09). This meta-analysis confirmed a high prevalence of EBV in the RA-LPD population. Furthermore, it provides evidence for the association between EBV presence and LPD susceptibility in RA patients, but not in those with other autoimmune diseases who developed LPD.

## 1. Introduction

Rheumatoid arthritis (RA) is a polygenic, multifactorial, and chronic inflammatory systemic autoimmune disease that affects up to 1% of the world’s population [1]. One of the well-known life threatening complications in RA patients is the development of lymphoproliferative disorders (LPDs), including malignant lymphoma. The evolution of LPDs in RA patients is complex and still enigmatic. Multiple risk factors have been identified, including RA-induced inflammation, infections, the use of immunosuppressive drugs for RA treatment, such as methotrexate (MTX), and a number of genetic factors [2]. The increased risk of developing malignant lymphoma in RA patients compared with the general population has been well documented [3]. The risk of Hodgkin lymphoma (HL)development, one of the most frequently identified LPDs, might be as much as 12 times higher in RA patients [4]. Although an earlier diagnosis of LPDs in RA patients favors better disease outcomes, their clinical and histopathological diversity represent a significant diagnostic challenge [5].

Here, it is clear that the hyperimmune state of RA itself and/or the immunosuppressive state induced by the administration of therapy might contribute to the evolution of LPDs. Pro-inflammatory cytokines, such as tumor necrosis factor alpha (TNF-α) and interleukin-6 (IL-6), have an essential role in RA pathogenesis [6]. Moreover, TNF-α operates as a growth factor for lymphoma and polymorphisms within TNF-α genes are associated with the higher risk of diffuse large B cell lymphoma (DLBCL), which is another frequent type of LPD in RA patients [7,8].

Epstein-Barr virus (EBV) infection is an important factor for LPDs development in RA patients [9]. In particular, a specific relationship that EBV establishes with the immune system has attracted attention. As a ubiquitous DNA herpesvirus that latently infects up to 99% of the world’s population, but with a complex mechanism of lifelong survival in cells, it represents a constant challenge to the host [10]. After the primary lytic infection of epithelial oropharyngeal and nasopharyngeal cells, EBV infects B lymphocytes, where it persists with the ability to exhibit occasional shifts between an active lytic cycle and latent state [11]. Mechanisms behind the role of this virus in RA could be based on several models of typical molecular mimicry in the initiation of RA [12]. In addition, the bystander damage surrounding EBV reactivation, the immortalization of B cells secreting self-reactive antibodies, and the resetting of the immune system favor more active overall immunity, resulting in reduced antigen tolerances [13]. The summarized, impaired control of EBV infection with cytokine activity leads to a reduced efficiency of peripheral blood T lymphocytes and expansion of EBV-infected B cells. Both of these events contribute to LPD development, with a 10-fold higher EBV load and high-titer antibodies to EBV antigens compared with controls [2,12].

Finally, the link between an infectious agent and the triggering of the autoimmune process has long been discussed. However, the current knowledge does not fully explain, not only the risk that the EBV infection carries for the development of LPDs in rheumatological patients, but also the role of this virus in the development of RA itself. Actually, it remains unclear whether the role of EBV is primarily in initiation of rheumatoid arthritis or in disease progression due to the chronic relapsing-remitting nature of EBV infection [14].

Understanding the influence of EBV infection in RA pathogenesis as an isolated causative agent or an accomplice in therapeutic risk is turning to be increasingly relevant. Considering the differences in design and patient population in previous studies, there is the lack of clear interpretation of obtained results and definitive conclusions regarding the association between EBV activity and RA complications, especially its contribution to LPD development. The aim of this systematic review and meta-analysis was to explore this problem.

## 2. Materials and Methods

This systematic review was performed in accordance with the PRISMA protocol (Reporting Items for Systematic Reviews and Meta-Analyses) and MOOSE guidelines for observational studies [15,16].

### 2.1. Study Selection

Publications were screened for inclusion in the systematic review in two phases. In addition, all of the disagreements were resolved by discussion at each stage with an inclusion of a third reviewer. Here, we included studies of all types of study design that detected the EBV virus in RA-LPD patients and any other group for comparison. Studies were eligible for inclusion if the EBV virus was detected in both groups. Studies were excluded if they: (i) Investigated other viruses;(ii) did not evaluate the presence of EBV in both groups: RA-LPD and control group; (iii) examined other populations (animal, cell lines); (iv) did not assess the presence of EBV, but its function; (v) were abstracts or (vi) were not original articles.

### 2.2. Database Search

Two biostatisticians with expertise in conducting systematic reviews and meta-analyses (AC, AB) developed the search strategy. A systematic review of peer-reviewed publications was performed through searches of three electronic databases: PubMed, Web of Science (WoS), and SCOPUS until 27 July 2021. Search queries and keywords for the PubMed search were: (Rheumatoid arthritis) and (lymphoproliferative disorder* or lymphoma or “EBV-Associated Lymphoproliferative Disorder*” or “Methotrexate-associated Lymphoproliferative Disorder*” or “Methotrexate-related Lymphoproliferative disorder*” or LPD or MTX-LPD or “Iatrogenic lymphoproliferative disorder*” or “Iatrogenic immunodeficiency-associated lymphoproliferative disorder*” or “Hodgkin-like lesion” or “reactive lymphoid hyperplasia” or “polymorphic-Lymphoproliferative disorder*” or PLD or “reactive lymphadenitis” or “Plasma cell myeloma”) and (Epstein Barr Virus or Epstein-Barr Virus or EBV or BURKITT’S LYMPHOMA VIRUS or Herpesvirus 4, human or HHV4); for WoS: ALL = (Rheumatoid arthritis)AND ALL = (lymphoproliferative disorder* or lymphoma or “EBV-Associated Lymphoproliferative Disorder*” or “Methotrexate-associated Lymphoproliferative Disorder*” or “Methotrexate-related Lymphoproliferative disorder*” or LPD or MTX-LPD or “Iatrogenic lymphoproliferative disorder*” or “Iatrogenic immunodeficiency-associated lymphoproliferative disorder*” or “Hodgkin-like lesion” or “reactive lymphoid hyperplasia” or “polymorphic-Lymphoproliferative disorder*” or PLD or “reactive lymphadenitis” or “Plasma cell myeloma”) AND ALL = (Epstein Barr Virus or Epstein-Barr Virus or EBV or BURKITT’S LYMPHOMA VIRUS or Herpesvirus 4, human or hhv8), and for SCOPUS: TITLE-ABS-KEY (“rheumatoid arthritis”) AND,TITLE-ABS-KEY (“lymphoproliferative disorder*”OR “lymphoma” OR “EBV-Associated Lymphoproliferative Disorder*” OR “Methotrexate-associated Lymphoproliferative Disorder*” OR “Methotrexate-related Lymphoproliferative disorder*” OR “LPD” OR “MTX-LPD” OR “Iatrogenic lymphoproliferative disorder*” OR “Iatrogenic immunodeficiency-associated lymphoproliferative disorder*” OR “Hodgkin-like lesion” OR “reactive lymphoid hyperplasia” OR “polymorphic-Lymphoproliferative disorder*” OR “reactive lymphadenitis” OR “Plasma cell myeloma”) AND TITLE-ABS-KEY (“EpsteinBarrVirus” OR “Epstein-BarrVirus” OR ebv OR “BURKITT’S LYMPHOMA VIRUS” OR “Herpesvirus 4, human” OR “HHV4”). Only publications in English were taken into account. In addition, reference lists of articles identified through electronic retrieval were manually searched, as well as relevant reviews and editorials. Experts in the field were contacted to identify other potentially relevant articles. Authors of relevant articles were contacted to obtain the missing data.

### 2.3. Article Screening and Selection

In the first step, two reviewers (A.C., D.M.) independently evaluated the eligibility of all of the titles and abstracts. Studies were included in the full text screening if either reviewer identified the study as potentially eligible or if the abstract and title did not include sufficient information for exclusion. Studies were eligible for full text screening if they included the detection of EBV virus in RA-LPD patients and control groups. According to the previously defined Inclusion and Exclusion criteria, in the second step, the same reviewers independently performed a full text screening to select articles for qualitative synthesis. Disagreements were resolved by consensus (A.C., D.M.) or arbitration (A.B., I.L.).

### 2.4. Data Abstraction and Quality Assessment

Two reviewers independently abstracted the following data: Author(s), country of research, year of publication, study design, study population, RA disease activity, specific type of LPD according to the WHO classification, LPD stage, age, gender, sample size, specimen for EBV detection, EBV positivity/negativity in cases and controls, method for EBV detection, and EBV latency. Each reviewer independently evaluated the quality of the selected manuscripts using an adapted version of the Newcastle-Ottawa tool for observational studies [17]. Reviewers used a standardized previously defined “EBV in RA-LPD protocol” when selecting and abstracting data. All of the detailed information regarding the reasons for inclusion/exclusion and quality assessment are available at https://osf.io/hb938/ (accessed on 5 December 2022).

### 2.5. Statistical Analysis

The primary outcome was the frequency of EBV positive patients in RA-LPD patients. As the outcome is dichotomous and the sample size varies, Mantel-Haenszel method was used as a measure of effect size to examine the differences in the ratio of EBV in the evaluated study groups from all of the primary articles. Mantel-Haenszel method is a fixed-effect meta-analysis method that uses a different weighing scheme that depends on which effect measure is used. Heterogeneity was assessed using the Chi-square Q test and I2 statistic. I2 presents the inconsistency between the study results and quantifies the proportion of observed dispersion that is real, i.e., due to between-study differences and not due to random error. The categorization of heterogeneity was based on the Cochrane Handbook [18] and states that I2 < 30%, 30% to 60% or >60%, corresponds to low, moderate, and high heterogeneity, respectively. Forest plots were constructed for each analysis showing the OR (box), 95% confidence interval (lines), and weight (size of box) for each trial. The overall effect size was represented by a diamond. Publication bias was assessed by the funnel plot for each defined outcome.

The meta-analysis of the prevalence was performed in order to estimate the prevalence of EBV positivity in the RA-LPD study group. The inverse variance method was applied. Data that were entered for each of the studies were the original prevalence from the study and the standard error of the prevalence according to the equation SQRT (*p** (1-*p*)/*n*), where *n* is the total number of respondents from the study.

Sensitivity analyses were conducted to examine the effects of removing the case-series from the analysis in the meta-analysis of the prevalence. The result was the same after the sensitivity analysis.

A *p*-value <0.05 was considered to be statistically significant. Analyses were performed using Cochrane Review Manager, version 5.4.

## 3. Results

### 3.1. Systematic Review

A total of 1294 potentially eligible articles were found. After the duplicates (*n* = 490) removed, the title and abstracts were evaluated for 804 articles. A total of 705 articles were excluded since they were not original articles, did not explore EBV, examined populations other than humans (animals, cell lines), did not evaluate RA-LPD patients or were in foreign languages. Of the 99 reviewed full text articles, 79 were selected for inclusion in the systematic review. A flow diagram illustrating this selection process is presented in Figure 1.

Characteristics of all 79 publications included in the systematic review are presented in detail in Appendix A. They were published between 1994 and 2021, with a total of 8653 participants; 3575 RA-LPD patients, and 5078 controls. The number of EBV positive RA-LPD patients was 1082, and there were 964 EBV positive controls. The minimum sample size of the RA-LPD and control groups was one. The maximum size of the RA-LPD group was 585, and the control group was 3187. There were six studies with the retrospective design (three case-control, one cross-sectional, and two nested case-control studies in the cohort study). Moreover, 13/79 studies were the descriptive case-series. Eight reported an unclear study design (six retrospective, one prospective and retrospective, and one retrospective observational). All of the other studies (52/79) did not report their study design. Most of the studies were from Japan (46). Others were from the USA (13), Sweden (4), France (4), Australia (2), Belgium (2), China (1), Denmark (1), and Korea (1). Four were multicenter studies. Diffuse large B cell lymphoma was the most frequent B cell LPD (54/139), and Classical Hodgkin’s lymphoma was the most frequent T cell LPD (51/98). Age ranged from 23 to 92 years in the RA-LPD group, and from 17 to 90 years in controls. The male vs. female ratio in the RA-LPD group was 1:2 in favor of females. The most frequently used antirheumatic drug was methotrexate, in 68/79 studies. The duration, week doses, and cumulative doses vary among the studies.

There were 172 case reports of patients who suffered from RA and developed LPD (Appendix A). The ratio between females and males was 2.6. The average age was 66.75 ± 10.98 years. The youngest patient was 24, while the oldest was 91 years. Almost all of the RA-LPD cases were EBV positive (72%). Moreover, 88% of these cases received methotrexate alone or in combination with other anti-rheumatic drugs.

### 3.2. Meta-Analysis of the Association between EBV Presence and LPD in RA Patients

According to the available data from the 79 studies included in the systematic review, we performed the meta-analysis of the prevalence in order to estimate the prevalence of EBV positivity in RA-LPD patients. The prevalence of a positive EBV result was high at 54% (OR = 1.54, 95% CI =1.45–1.64, *p* < 0.001) (Figure 2).

Thirty-one studies with the available EBV data were included in the meta-analysis of the association between EBV presence in RA-LPD and controls. Control groups were heterogeneous and included patients with LPD only, patients with autoimmune diseases other than RA and LPD, patients with systemic rheumatic diseases other than RA with and without LPD, and immunosuppressed patients without RA. There was a significant association between the positive EBV result and LPD susceptibility in RA patients when compared with all of the designated controls (OR = 1.88, 95% CI = 1.29–2.73, *p* = 0.001) (Figure 3). As the control groups were heterogeneous, the meta-analysis of the association between EBV presence in RA-LPD and specific control groups was performed as a subgroup analysis. There was a significant association between the positive EBV result and LPD susceptibility in RA patients when compared with LPD controls (OR = 1.92, 95% CI = 1.15–3.19, *p* = 0.010) (Figure 4). On the contrary, there was no association between EBV positivity in RA-LPD patients when compared with patients who suffered from autoimmune diseases other than RA and LPD (OR = 0.79, 95% CI = 0.30–2.09, *p* = 0.630) (Figure 5).

### 3.3. Regional Analysis

The prevalence of an EBV positive result in RA-LPD patients was evaluated in relation to the continent of origin (America, Asia, and Europe). The highest prevalence was in Asia, 65% (OR = 1.65, 95% CI = 1.55–1.76, *p* < 0.001), followed by North America, 39% (OR = 1.39, 95% CI = 1.25–1.54, *p* < 0.001), whereas the lowest prevalence was seen in Europe, 22% (OR = 1.22, 95% CI = 1.13–1.31, *p* < 0.001) (Figure 6). Moreover, the meta-analysis of the association between EBV presence in RA-LPD and all of the designated controls was performed in relation to the continent of origin (America, Asia). It was shown that there is a significant association between the positive EBV result and LPD susceptibility in RA patients when compared with all of the controls in Asia (OR = 2.93, 95% CI = 1.89–4.55, *p* < 0.001), but not in North America (OR = 0.87, 95% CI = 0.39–1.95, *p* = 0.730) (Figure 7).

## 4. Discussion

This study provided the first systematic review with a meta-analysis that confirmed a high prevalence of EBV in the RA-LPD population (54%). Additionally, this meta-analysis has shown the association between a positive EBV result and LPD susceptibility in RA patients when compared with all of the controls or when compared with the LPD controls separately, but did not show this association when compared with controls defined as patients with autoimmune diseases other than RA who developed LPD.

In the first place, the etiology of LPD includes a still insufficiently defined genetic predisposition, particularly the role of the human leukocyte antigen (HLA). Although, for example, Genomewide association studies (GWAS) have demonstrated a link between HLA class II and Hodgkin’s lymphoma, it was suggested that EBV-positive and EBV-negative Hodgkin’s lymphoma have a different genetic susceptibility. HLA class I alleles are associated with EBV-positive, and HLA class II alleles with EBV-negative Hodgkin’s lymphoma [9]. In addition to the intertwining of genetic background and the influence of EBV infection for lymphoma development in general, there are scarce reports linking the genetic susceptibility of patients with RA to LPD. For example, the association of HLA-B15:11 and EBV-positive RA-LPD was reported [19].

During the life-long persistence in memory B cells, EBV remains largely limited in its activity and replication capacity. This unique property of its viral life cycle is based on the ability to express a different set of latent genes in three latency programs: Six genes of EBV nuclear proteins (EBNA-1, -2, -3A, -3B, -3C, -LP), three genes of latent membrane proteins (LMP-1, -2A, -2B), and two non-coding RNAs (EBER-1, -2). Each of the programs could lead to cell transformation and the development of specific malignancies. Moreover, EBV-associated diseases, and consequently each EBV-positive malignancy is clearly defined by the set of expressed genes that characterize one of three latency programs: Burkitt‘s lymphoma has a type I latency pattern (EBER+, LMP1-, and EBNA2-); Hodgkin lymphoma and a variety of non-Hodgkin lymphomas have a type II pattern (EBER+, LMP1+, and EBNA2-); and post-transplant lymphoproliferative disorders (PTLD) that develop in an immunocompromised host most often have the type III latency pattern (EBER+, LMP1+, and EBNA2+) [20]. This last category of LPDs (PTLD) is not the most prevalent among RA patients. However, it is most commonly positive for EBV due to the absence of effective T cell surveillance on the one hand, and virus-transformed B cell proliferation on the other hand [9]. According to the literature data, RA-LPDs are more likely to be latency type II or latency type II followed by latency type III, which indicates mild immunosuppression [19,21,22]. Taking into account multiple distinct pathogenetic mechanisms of LPDs and their different associations with the EBV infection, additional meta-analyses that deal with this problem are separately required for each entity.

According to the studies included in this analysis, it is observed that there is a lack of common criteria for determining the EBV presence or a positive result in clinical specimens used in the mentioned studies. Immunohistochemistry and in situ hybridization were the most commonly used methods for the detection of EBER or LMP1. However, limitations for interpreting and understanding the viral replication capacity are based on a wide range of cut-offs of some of the methods used, using different methods even within the same studies, and finally, the use of indirect serology testing, such as ELISA [23].

Among the risk factors for LPD in RA patients that are listed as possible triggers during the EBV infection are: the association between the infection and gene promoter methylation [24]. Actually, a hypothetical mechanism with low Bcl-2 expression and low hypermethylation of apoptosis-related genes may accelerate the apoptosis of LDP cells and may explain the massive necrosis that is observed in the histological examination and spontaneous rapid regression of EBV-positive LPD after MTX withdrawal [9]. Although the aim of our study did not address the impact of therapy on LPD development, the extracted data showed that the most frequently used antirheumatic drug was methotrexate in 86% of studies. Therefore, one of the essential ambiguities that is important for the establishment of therapeutic protocols for RA-LPD patients still remains as the effect of MTX and the most commonly used drugs in the treatment of RA on the regulation of EBV genes. The quick regression of EBV-positive LPD patients with RA who are treated with MTX after drug withdrawal, suggests that MTX could have a possible influence on the regulation of EBV replication, which was also demonstrated in vitro [9,25]. On the other hand, experiments that measured the EBV load or EBV-specific Tcell responses in RA patients before and after short-term exposure to MTX or TNF inhibitor did not find a significant change in either of them [26].

A considerable number of publications showed individual cases of LPD development in RA patients. As many as 172 case reports were found from this systematic review. Although these cases could not be processed by the meta-analysis, they undoubtedly indicate the association between EBV presence and LPD susceptibility in RA patients with a higher prevalence of EBV positive results in RA-LPD cases than in the systematic review (72% vs. 54%). An almost identical percentage of patients from the case reports received MTX alone or in combination with other anti-rheumatic drugs compared withthe patients included in the meta-analysis.

A geographic variation of an EBV prevalence in patients with different EBV-associated disorders has been well documented in previous studies [23]. The geographic heterogeneity of an EBV prevalence in RA-LPD patients with the highest prevalence in Asia, followed by North America and Europe, which our study confirmed, is consistent with the previously reported data [27,28,29]. In addition, the meta-analysis of the association between EBV presence in RA-LPD and controls in relation to the continent of origin showed that there was a significant association between a positive EBV result and LPD susceptibility in RA patients when compared with all controls in Asia, but not in North America. Various theories have been proposed that explain the geographic differences in the prevalence of EBV-associated disorders: Different distributions of EBV strains and geographically associated EBV gene polymorphisms; host genetic predisposition; interaction between environmental carcinogens; and differences in the age of primary infection that affect the incidence of EBV-manifested diseases as notable examples [23,30,31].

A significant result of this study that should also be pointed out is the absence of the association between a positive EBV result and LPD susceptibility in RA patients when compared with controls, which is defined as patients with autoimmune diseases other than RA who developed LPD. These data support earlier theories regarding the role of EBV infection in a broad spectrum of autoimmune diseases [14,32]. Our control group defined as “autoimmune diseases other than RA” has mainly consisted of patients with systemic autoimmune diseases (SADs), a group of partially overlapping syndromes, also called connective tissue diseases, since they are often accompanied by inflammation of connective tissues. The SADs include rheumatoid arthritis (RA), Sjögren’s syndrome (SS), systemic lupus erythematosus (SLE), systemic scleroderma (SSc), etc. For all these different diseases, there is an increased tendency to develop cancer, including various forms of lymphoma [14].

Our study has several limitations. They originate from the control group variability, a significant number of case reports, huge numbers of different LPD diagnoses within case groups, poor design of original articles, and difficulty to analyze the effect of different factors to the previously assessed effect size. Heterogeneous control groups (rheumatology diseases, autoimmune diseases, healthy respondents, etc.) may lead to the under/over estimation of the real effect of EBV positive results to the LPD susceptibility in RA patients. A large pool of case reports represents a good descriptive base for further studies. However, 172 case reports screened through our systematic review undoubtedly indicate the association between EBV presence and LPD susceptibility in RA patients. On the contrary, more accurate and valuable results would be obtained from our meta-analysis if these studies could have been used. It is important to highlight the significance of non-homogeneous case groups, as well. The advantage of using a specific LPD diagnosis in RA patients will allow clear conclusions. Although a prospective study design allows a better quality of evidence, most of the studies from this systematic review were of retrospective design. Almost all of the included studies evaluated RA-LPD patients and controls that were treated with MTX, which resulted in homogeneous groups according to this characteristic. This is the reason why it was impossible to evaluate the effect of MTX use to the previously calculated probability of LPD susceptibility in RA-LPD patients and controls through meta-regression.

Despite the growth in the number of new therapeutic options that aim to achieve optimal inflammation control and minimize or even prevent the key complications of RA, MTX is still the most commonly used drug for RA-LPDs. In addition, EBV is the most frequently described infectious agent in relation to LPD development in RA patients. Moreover, there are still no distinct features that distinguish MTX-LPDs from other LPDs in RA patients described in the published reports. Guidelines, that are more than necessary for the treatment of RA-LPDs, have not yet been established due to the unknown underlying mechanism of pathogenesis and the baseline risk of malignancies in patients with RA. Therefore, the question of how EBV can be involved in an array of diverse diseases, and how so many, apparently diverse, autoimmune diseases could have a similar risk for the development of LPD, brings up the necessity for a more specific view. For example, earlier literature suggestions, such as the histological categorization or analysis of EBV EBER positivity should be taken into account in the further diagnosis and categorization of LPD patients, i.e., disease course prediction and modulation strategy of existing therapeutic protocols [33].

## 5. Conclusions

This systematic review with meta-analysis confirmed a high prevalence of EBV in the RA-LPD population, which points to the role of EBV in the pathogenesis of this complication. However, for the first time, one meta-analysis has shown that there is no association between EBV positivity and LPD susceptibility in RA patients when comparing patients with autoimmune diseases other than RA who developed LPD. This observation shows that EBV plays a distinct role in the pathogenesis of LPDs in RA patients, but also suggests that EBV may contribute to the development of LPDs in other autoimmune diseases.

## Figures and Tables

**Figure 1 viruses-14-00115-f001:**
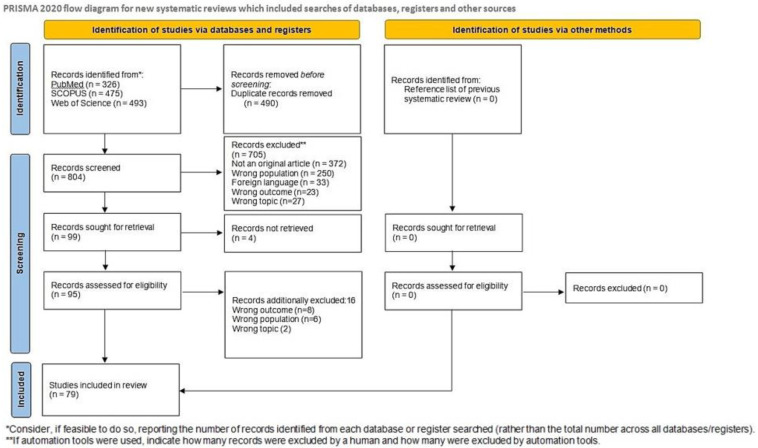
Flow diagram.

**Figure 2 viruses-14-00115-f002:**
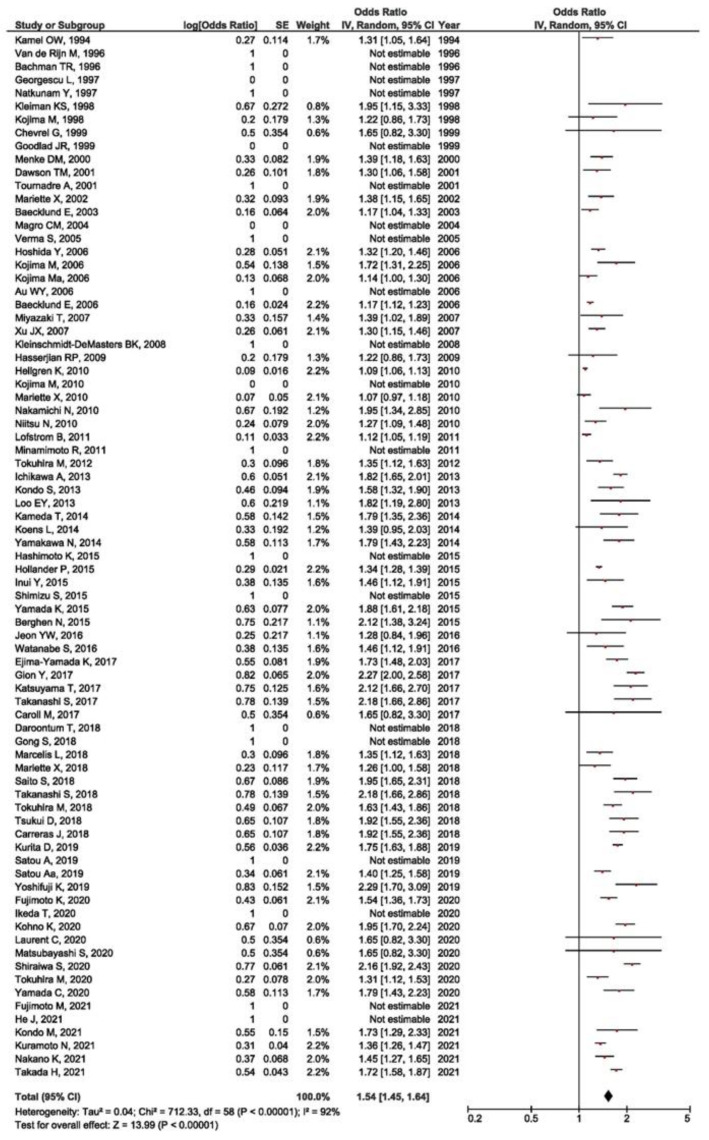
Meta-analysis of the prevalence of positive EBV result in RA-LPD patients (red dot represents the point estimate—prevalence and black frame represents 95%CI of the point estimate for the data from each of included studies).

**Figure 3 viruses-14-00115-f003:**
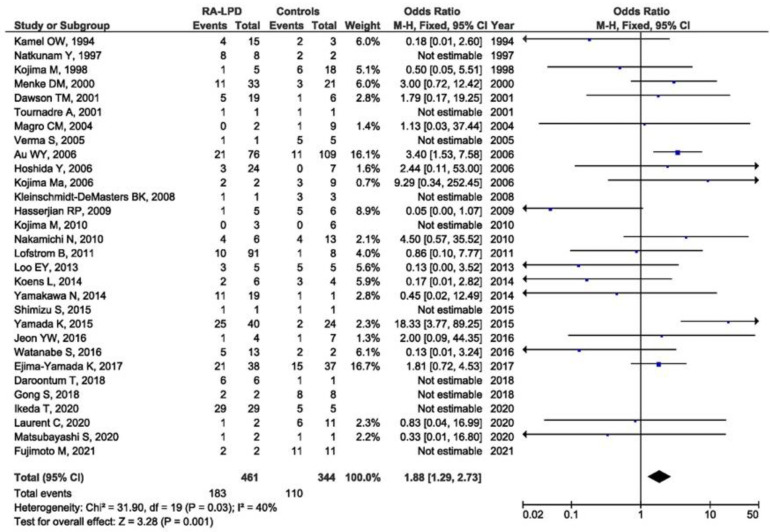
Meta-analysis of the difference in the positive EBV result in RA-LPD patients and all controls (blue dot represent the odds ratio and black frame 95% CI of the odds ratio from each of the included studies. black arrows show that upper and/or lower limit of the 95%CI of odds ratio is below or above defined values on the x axis).

**Figure 4 viruses-14-00115-f004:**
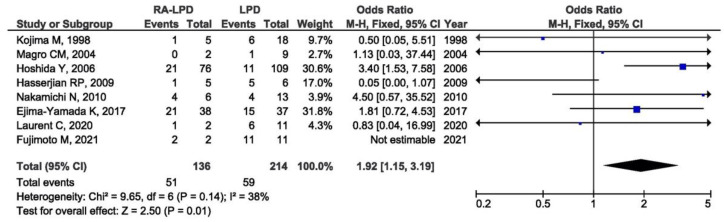
Meta-analysis of the difference in the positive EBV result in RA-LPD and LPD patients (blue dot represent the odds ratio and black frame 95% CI of the odds ratio from each of the included studies. black arrows show that upper and/or lower limit of the 95%CI of odds ratio is below or above defined values on the x axis).

**Figure 5 viruses-14-00115-f005:**
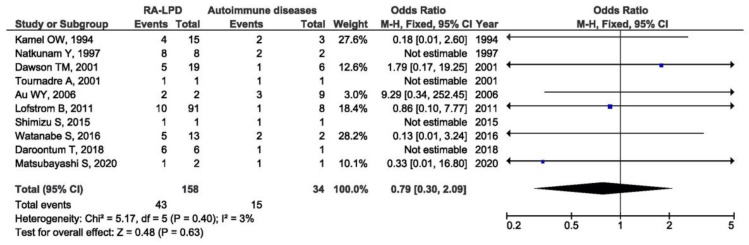
Meta-analysis of the difference in the positive EBV result in RA-LPD patients and patients with autoimmune diseases other than RA and LPD (blue dot represent the odds ratio and black frame 95% CI of the odds ratio from each of the included studies. black arrows show that upper and/or lower limit of the 95%CI of odds ratio is below or above defined values on the x axis).

**Figure 6 viruses-14-00115-f006:**
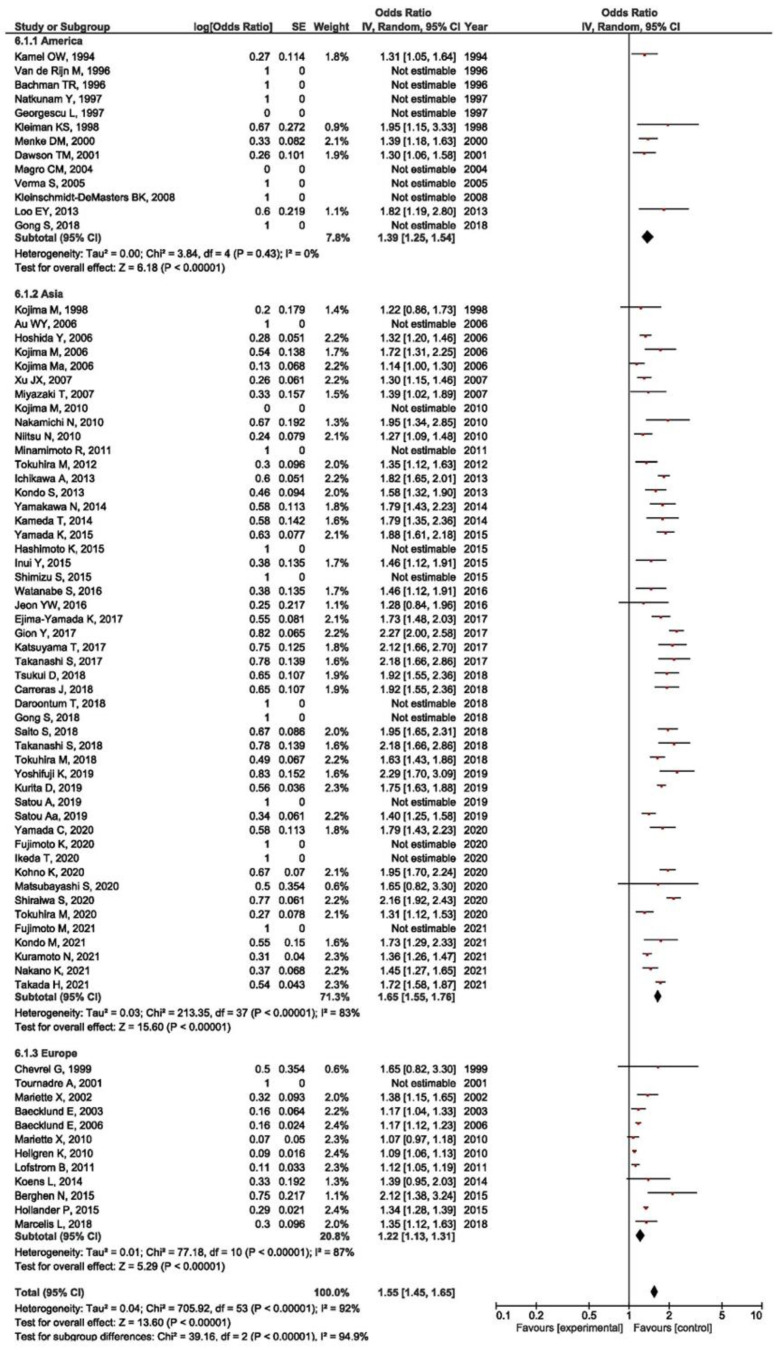
Meta-analysis of the prevalence of the positive EBV result in RA-LPD patients by the continent of origin (red dot represents the point estimate—prevalence and black frame represents 95%CI of the point estimate for the data from each of included studies).

**Figure 7 viruses-14-00115-f007:**
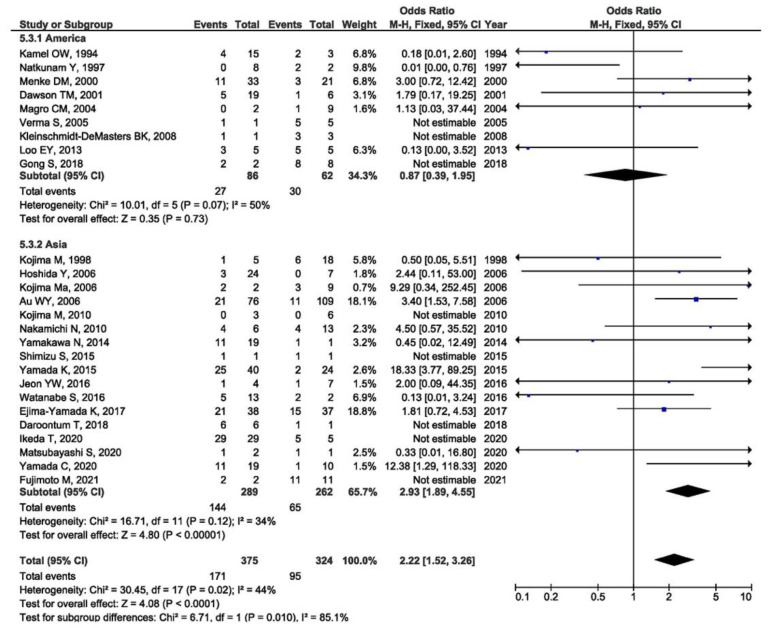
Meta-analysis of the difference in the positive EBV result in RA-LPD patients and all controls by the continent of origin (blue dot represent the odds ratio and black frame 95% CI of the odds ratio from each of the included studies. black arrows show that upper and/or lower limit of the 95%CI of odds ratio is below or above defined values on the x axis).

## Data Availability

All of the detailed information regarding the reasons for inclusion/exclusion and quality assessment, as well as Appendix A are available at https://osf.io/hb938/ (accessed on 5 December 2021).

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
