# Peer review of "New Evidence of Significant Association between EBV Presence and Lymphoproliferative Disorders Susceptibility in Patients with Rheumatoid Arthritis: A Systematic Review with Meta-Analysis"

_viruses, 2022, doi:10.3390/v14010115_

Round 1

Reviewer 1 Report

This work reports interesting evidence for the association between Epstein–Barr virus infection and lymphoproliferative disorders susceptibility in patients with rheumatoid arthritis. It is the first systematic review with meta-analysis that confirmed this association.

This is an interesting and robust systematic review with engaging content. I do not feel that I could suggest any major thing which may improve the manuscript and for that reason I am recommending acceptance in its current form. The only minor thing which I can suggest is moving Table 1 to supplementary data because it presents large amounts of data in 19 pages, and thus it may annoy readers. Also, please check page numbering.

Author Response

According to the suggestions we moved Table 1 to supplementary data and corrected page numbering.

Reviewer 2 Report

1: Since a hypothesis or a clear initial question was not stated in the manuscript and no uniquely generated y clinical trial data was used during meta-analysis, the authors should revise the title and remove meta-analysis. They should also remove meta-analysis from the text where appropriate. For example: Considering this unresolved issue, we performed a designed this systematic review and meta-analysis with the aim to explore the link between EBV infection and LPD risk in patient with RA. Lines 83-85.

2: It is unclear from the literature review if abstracted data from their publication set may have introduced  “carryover/duplicate information” from referenced authors subsequent papers. The authors might state clearly in the methods section that evidence of duplicate data was excluded, and this action was initiated during the pre-analysis stage. Failure to account for duplicate data could introduce data bias.

3: As pointed out correctly in the authors in their study limitations differences in LPD diagnoses (classification) may lead to under or over estimation concerning the impact of EBV positivity and LPD susceptibility in RA patients. The authors use a broad classification of LPD that include or do not include EBV association. For example, Burkitt’s lymphoma included African Burkitt’s,  that is strongly associated with EBV and sporadic Burkitt’s lymphoma that is not associated with EBV. Similarly, Hodgkin's lymphomas are not commonly associated with EBV as the actual Reed Sternberg is lacking EBV markers, yet the invasive B cells associated with the tumor contain EBV . Based on EBV serology 90 to 95% of individuals are infected with EBV by the age of thirty. It is still unclear whether EBV simply exploits bystander cell targets and not acting as a primary cause and progenitor of the cancer cell.

This casting of a “broad net classification of LPD ” when gathering data for EBV associated lymphomas and RA may have inclusion positive samples versus compared to the matching control population. This is broad classification most exemplified by the authors observing no difference in RA incidence of LPD to that of autoimmune patient. (Lines 338-347) . The authors should more fully discuss this important finding.

4: The authors could  expound on the role of methotrexate and cancers. It is well documented in the literature that  lowering immunosuppressives in organ transplant recipients leads to regression or reversal of EBV lymphoproliferative disease. Therefore, it is the level of immunosuppression that is a major factor in EBV lymphomas and lymphoproliferative diseases.

Author Response

1. In Materials and Methods section, subsection Study selection, we added the explanation “We included studies of all types of study design that detected EBV in RA-LPD patients and any other group for comparison”.

We kindly recommend to leave “meta-analysis” in the title of our article. In order to explain this recommendation, we enclose a definition of meta-analysis that states: "Meta-analysis is a quantitative, formal, epidemiological study design used to systematically assess previous research studies to derive conclusions about that body of research (Haidich, 2010)". Also, meta-analysis must contain systematic review, so the term meta-analysis is broader than systematic review, and our article is meta-analysis with systematic review in itself.

Haidich AB. Meta-analysis in medical research. Hippokratia. 2010 Dec;14(Suppl 1):29-37. PMID: 21487488; PMCID: PMC3049418.

2. Given the possibility of duplication of data, we were very careful not to include studies whose subjects partially or completely overlapped.

3. In this meta analysis, we have presented a series of distinct LPDs associated with EBV infections, just as presented by the authors of the articles included in our meta analysis. We agree that the outcome might have been slightly different if the meta analyzes could have been done for each LPD individually. However, after extraction of the study data that were processed, it was noticed that there is insufficient data for specific diagnosis to meet the authors aims but also the above request of the reviewer. On the other hand that was not the main focus of the meta analysis. We would also like to mention that regardless of the indisputable different association between EBV and different LPDs, complexity in pathogenesis is contained in the genetic basis, other external triggers, immune status and in various EBV latency programs, too (already discussed in lines 288-301). It is impossible to process all these aspects in one publication and for the mentioned reasons it requires several separate analyzes. However, in order to emphasize this, accepting the reviewer's suggestions, we added this observation to the discussion as a new sentence: “Taking into account multiple distinct pathogenetic mechanisms of LPDs and their different association with EBV infection, additional meta-analyzes dealing with this problem are required for each entity separately.” (lines 301-303).

Going back to the beginning and setting of this meta-analysis, the initial common feature of all patients in this study is rheumatoid arthritis with its potential specificities it has on the immune status, the consequences of infections, described mimicry with EBV, inability to maintain control of latent infections, inflammatory processes, etc. Unlike the limitations mentioned for the LPD classification, analyzing the association between EBV presence and LPDs susceptibility in patients with RA, we got comprehensive data so we could perform some additional analyzes that were feasible on the basis of data availability. Were performed: RA-LPD vs. LPD patients, RA-LPD vs. autoimmune diseases other than RA with LPD patients, as well as geographical differences in LPD susceptibility between RA-LPD and all designated controls.

From all the above, we suggest that in addition to the existing parts of the discussion dedicated to the described topics (lines 276-283, 284-301, 335-348, 349-360), new sentence (lines 301-303), as well as the more detailed limitations of the study (lines 361-379), we do not further deepen the discussion. There is a risk that considering all aspects of the pathogenesis of LPDs separately, autoimmunity and the latency phases of EBV infection the discussion will become too long and difficult to read.

4. The aim of our study did not address the impact of therapy on LPD development, because extracted data showed that almost all evaluated studies in cases, as well as in controls, used methotrexate (86% of studies). Respecting the reviewer's suggestion, we have further noted this in the study limitations: ”Almost all included studies evaluated RA-LPD patients and controls treated with MTX that gave homogeneous groups according to this characteristic. That is why it was impossible to evaluate the effect of MTX use to previously calculated probability of LPD susceptibility in RA-LPD patients and controls through meta-regression.”